



Assessing contributions of natural surface and anthropogenic emissions to
atmospheric mercury in a fast developing region of Eastern China from
2015 to 2018
Xiaofei Qin[1], Leiming Zhang[2], Guochen Wang[1], Xiaohao Wang[3], Qingyan Fu[3], Jian Xu[1], Hao Li[1],
Jia Chen[1], Qianbiao Zhao[3], Yanfen Lin[3], Juntao Huo[3], Fengwen Wang[4], Kan Huang[1,5,6,*], Congrui
Deng[1,*]
[1] Center for Atmospheric Chemistry Study, Shanghai Key Laboratory of Atmospheric Particle
Pollution and Prevention (LAP3), Department of Environmental Science and Engineering, Fudan
University, Shanghai, 200433, China
[2] Air Quality Research Division, Science and Technology Branch, Environment and Climate Change
Canada, Toronto, M3H 5T4, Canada
[3] Shanghai Environmental Monitoring Center, Shanghai, 200030, China
[4] State Key Laboratory of Coal Mine Disaster Dynamics and Control, College of Environment and
Ecology, Chongqing University, Chongqing 400030, China
[5] Institute of Eco-Chongming (IEC), Shanghai, 202162, China
[6] Institute of Atmospheric Sciences, Fudan University, Shanghai 200433, China
**Abstract**
Mercury (Hg) is a global toxic pollutant that can be released into the atmosphere through
anthropogenic and natural sources. The uncertainties in the estimated emission amounts are much
larger from natural than anthropogenic sources. A method was developed in the present study to
quantify the contributions of natural surface mercury emissions to ambient gaseous elemental
mercury (GEM) concentrations through application of positive matrix factorization (PMF) analysis
with temperature, $O_3$, and $NH_3$ as indicators of GEM emissions from natural surfaces. GEM
concentrations were continuously monitored at a 2-hourly resolution at a regional background site
in the Yangtze River Delta in Eastern China during 2015-2018. Annual average GEM concentrations
were in the range of 2.03-3.01 ng/m$^3$, with a strong decreasing trend at a rate of -0.32±0.07 ng m$^{-3}$
yr$^{-1}$ from 2015 to 2018, which was mostly caused by reduced anthropogenic emissions since 2013.
The estimated contributions from natural surface emissions of mercury to the ambient GEM
concentrations were in the range of 0.90-1.01 ng/m$^3$ on annual average with insignificant interannual
changes, but the relative contribution increased significantly from 36% in 2015 to 53% in 2018,
gradually surpassing those from anthropogenic sources.

**1. Introduction**





Mercury has long been recognized as a toxic pollutant due to its bioaccumulation and health
effects (Driscoll et al., 2013b;Clarkson and Magos, 2006;Schroeder and Munthe, 1998;Horowitz et
al., 2017;Fu et al., 2012;Wright et al., 2018). Mercury in the atmosphere can be transported globally,
mostly in the form of gaseous elemental mercury (GEM) due to its long lifetime in air (Driscoll et
al., 2013a). Clarifying sources and quantifying emissions from the major sources of atmospheric
mercury are critical for understanding the biogeochemical cycle of mercury and developing mercury
reduction strategies. Mercury in the atmosphere is released from both natural and anthropogenic
sources. Natural sources include volcanoes. geological weathering, forest fires, re-emissions of pre-
deposited mercury from natural surfaces, etc (Gustin et al., 2008;Mason and Sheu, 2002). Among
these sources, emissions from natural surfaces are the major ones and a number of studies have been
devoted to understanding the processes of natural surface emissions (Xu et al., 1999;Lindberg et al.,
2002;Kocman et al., 2013). Anthropogenic sources mainly include coal-fired power plants, non-
ferrous smelters, and waste incineration (Friedli et al., 2009). Globally, natural sources released
about 5200 tons mercury into the atmosphere on an annual basis, which contributed up to two-thirds
of the global atmospheric mercury budget, while those by anthropogenic sources was estimated to
be around 2300 tons (Pirrone et al., 2010). In China, the total mercury emissions released from
natural and anthropogenic sources were estimated to be 574.5 ton yr$^{-1}$, and 571 ton yr$^{-1}$, respectively
(Wang et al., 2016;Zhang et al., 2015).
During the past decades, anthropogenic emissions of mercury in Europe and North America
have been reduced significantly through phasing out mercury from many commercial products as
well as benefiting from $SO_2$ and $NO_x$ emission reduction from coal-fired utilities, resulting in
considerable decrease in atmospheric mercury concentrations in these regions (e.g., approximately
1-2% yr$^{-1}$ decrease from 1990 to 2013) (Streets et al., 2011;Zhang et al., 2016). In China,
anthropogenic mercury emissions decreased from 571 ton in 2013 to 444 ton in 2017 due to the co-
benefits of aggressive air pollutant control measures implemented in this period (Liu et al., 2019a).
GEM concentrations measured at a rural site north of Shanghai showed a substantially decreasing
trend from 2014 to 2016 (Tang et al., 2018).
With the decrease of anthropogenic mercury emissions in many parts of the world (Zhang et
al., 2016), the contributions of natural emissions to total mercury budget are expected to be more
important. However, the trends of natural emissions are still unclear due to the difficulties in directly



measuring GEM emissions from natural surfaces (Zhu et al., 2015). Existing estimates of GEM
emission from natural sources have large uncertainties (e.g., from 1500 to 5207 Mg yr$^{-1}$ on global
scale), limiting our understanding of the role of natural emissions in the global mercury cycle (Song
et al., 2015;Wang et al., 2014b). For example, a study at rural Beijing showed that modeled GEM
concentrations were underestimated by about 40% than measurements from April to September
2009 due to the absence of natural emission inventories (Wang et al., 2014a). Hence, it is meaningful
to develop a method to quantify the contributions of natural surface emissions to total mercury
budget in the atmosphere, especially in China where anthropogenic emissions have been fast
decreasing in recent years.
The purpose of the present study is to differentiate the contributions of natural surface
emissions and anthropogenic emissions to the measured ambient GEM concentrations collected
during a four-year period at a regional background site in the Yangtze River Delta (YRD) of Eastern
China. This was done by conducting positive matrix factorization (PMF) analysis with identified
variables as tracers of natural surface mercury emissions. Results presented in this study provide an
approach that can be potentially used for improving mercury emission databases for natural sources.

**2. Materials and methods**
**2.1 Site description**
Shanghai, situated in the YRD region, is one of the most developed cities in China. Like in
many other cities in China, severe air pollutions have occurred frequently in this city in the past
decades. A supersite has been set up next to the Dianshan Lake in Qingpu District of rural Shanghai
(Figure 1) as part of the framework of State Environmental Protection Scientific Observation and
Research Station. This supersite is designed to represent the regional scale air pollution
characteristics in the YRD region based on the following two considerations: (1) it is located in the
conjunction area of Shanghai, Jiangsu, and Zhejiang provinces; and (2) there are no large point
sources such as coal-fired power plants, nonferrous metal smelting, and cement production within
20km distance surrounding the site. This site was established in 2013 and its capacity has been
gradually built by measuring a set of atmospheric parameters, including meteorological factors,
trace gases, aerosol physical and chemical parameters, vertical profiles of ozone and particles, etc.
More detailed descriptions of the site can be found elsewhere (Qin et al., 2019;Duan et al., 2017).




## 2.2 Measurements of gaseous elemental mercury


An automated mercury vapor analyzer Tekran 2537B/1130/1135 was installed on the third floor
of a building for real time continuous GEM measurements since January 2015. GEM was measured
based on the principle of cold vapor atomic fluorescence spectroscopy (CVAFS) (Landis and Keeler,
2002). Briefly, ambient GEM was collected on gold traps and then thermally decomposed to GEM
before detection. The sampling interval of GEM was 5 minutes with a flow rate of 1L/min. More
details of this instrument can be found elsewhere (Mao et al., 2008).
Strict quality control procedures were followed during the sampling process. Denuders and
quartz filters were prepared and cleaned according to the instructions in Tekran technical notes
before sampling. Routine calibration with internal permeation source was performed every 47 hours
and manual injections of standard saturated mercury vapor were conducted to ensure the accuracy
of these automated calibrations. The KCl-coated denuder, Teflon-coated glass inlet, and impactor
plate were replaced weekly and quartz filters were replaced monthly. Individual extremely high
GEM concentrations that occasionally happened were regarded as outliers and were excluded from
the data analysis.

## 2.3 Measurements of other air pollutants and meteorological parameters


Water soluble ions in $PM_{2.5}$ and soluble gases were continuously measured by Monitor for
Aerosols and Gases in ambient Air (MARGA) operated at a flow rate of 16.7 L/min with a time
resolution of one hour, as detailed in (Chang et al., 2016). Briefly, water-soluble gases in the airflow
were removed by an absorbing liquid, then the particles were induced by a supersaturation of water
vapor to grow into droplets before they were collected and transported into the analytical chamber.
Trace metals in $PM_{2.5}$ were continuously measured by using the Xact 625 ambient metals
monitor (Cooper Environmental, Beaverton, OR, USA) operated at a flow rate of 16.7 L/min with
hourly resolution, as detailed in (Yu et al., 2019). Briefly, the particles in the airflow were deposited
onto a Teflon filter tape, and then transported into the spectrometer where the particles were
analyzed with an X-ray fluorescence.
Ozone, carbon monoxide, and $PM_{2.5}$ were measured by Thermo Fisher 49i, Thermo Fisher 48i-
TLE, and Thermo Fisher 1405-F, respectively. Meteorological parameters including ambient


temperature, wind speed, and wind direction were obtained at the sampling site by using the
automatic weather station (AWS). Bivariate polar plots (BPP) were applied in this study to explore
how GEM concentrations change with different wind direction and wind speed, which has proven
to be a reliable method for identifying different source regions (Carslaw et al., 2006;Carslaw and
Ropkins, 2012;Chang et al., 2017). Here, the open-source software "openair" in R was used to create
BPPs (Carslaw and Ropkins, 2012).

**2.4 Positive matrix factorization (PMF)**
The PMF model has been proven to be a useful tool to provide quantitative source profiles and
source contributions (Xu et al., 2017;Gibson et al., 2015). The basic principle of PMF is that
concentrations of the samples were determined by the source profiles with different contributions,
which can be descried as follows:
$$X_{ij} = \sum_{k=1}^{P} g_{ik} f_{kj} + e_{ij} \quad (1)$$
where $X_{ij}$ represents the concentration of the $j$th species in the $i$th sample, $g_{ik}$ is the contribution
of the $k$th factor in the $i$th sample, $f_{kj}$ provides the information of the mass fraction of the $j$th
species in the $k$th factor, $e_{ij}$ is the residual for specific measurement, and $P$ represents the number
of factors.
The objective function expressed in Eq. (2) below, which is the sum of the square of the
difference between the measured and modeled concentrations weighted by the concentration
uncertainties, needs to be minimized before the PMF model determines the optimal non-negative
factor profiles and contributions.(Cheng et al., 2015)
$$Q = \sum_{i=1}^{n} \sum_{j=1}^{m} \left( \frac{X_{ij} - \sum_{k=1}^{p} A_{ik} F_{kj}}{S_{ij}} \right)^2 (2)$$
where $X_{ij}$ represents the concentration of the $j$th contamination in the $i$th sample, $m$ is the total
number of pollutant, and $n$ is the total number of sample. $A_{ik}$ represents the contribution of the $k$th
factor on the $i$th sample and $F_{kj}$ represents the mass fraction of the $j$th pollutant in the $k$th factor.
$S_{ij}$ is the uncertainty of the $j$th pollutant on the $i$th factor and $P$ is the number of factors. In this
study, we explored the number of factors from three to eight with the optimal solutions determined
by the slope of the Q value versus the number of factors. For each run, the stability and reliability
of the outputs were assessed by referring to the Q value, residual analysis, and correlation
coefficients between observed and predicted concentrations. Finally, we found that a six-factor



solution showed the most stable results and gave the most reasonable interpretation. A dataset
containing uncertainty values of each species was created and digested into the model, with the error
fraction being assumed to be 15% for GEM concentration and 10% for other compounds (Xu et al.,
2017;Polissar et al., 1998).
It should be noted that Fpeak model run at the strength of 0.5 was done by using the rotation
tools in PMF and the results were summarized in Table S1. For all seasons, the increase of the Q-
value due to the Fpeak rotation with a dQ was less than 1% of the Base Run Q (robust) value.
According to the User Guide of PMF5.0, it was acceptable when the %dQ was less than 5%. The
profiles and contributions of each source were examined and there were no significant differences
between the factor contributions of Base Run and rotation results. Hence, the Base Run results were
used in this study.

**2.5 Annual changes of anthropogenic mercury emission in China and YRD**
It was reported that the annual anthropogenic atmospheric mercury emission in China
significantly increased from 147 tons in 1978 to 549 tons in 2010 (Wu et al., 2016). In more recent
years, in order to cope with the severe air pollution situation, the Chinese government has taken
many rigorous and ambitious measures such as introduction of ultra-low emissions standards on
power plants and phasing out of small factories with high-emissions (Zheng et al., 2018). As a result,
mercury emissions from anthropogenic sources have since been declining in China. For the five-
year period of 2013-2017, annual total anthropogenic mercury emissions in China were estimated
to be 571, 547, 528, 486, and 444 tons, respectively, or a total decline of 127 tons. During the same
period, the reduction of anthropogenic mercury emissions reached 60 tons in eastern China (Liu et
al., 2019a).

**3. Results and Discussion**
**3.1 The measured gaseous elemental mercury**
**3.1.1 Decreasing trend of gaseous elemental mercury**
The measured annual mean GEM concentrations were $3.01 \pm 1.03$, $2.58 \pm 0.84$, $2.52 \pm 0.84$,
and $2.03 \pm 0.69$ ng/m$^3$ from 2015 to 2018. By using the Theil-Sen function, monthly GEM exhibited
a significantly decreasing trend from 2015 to 2018 ($p<0.05$) with a rate of $-0.32 \pm 0.07$ ng m$^{-3}$ yr$^{-1}$





(Figure 2a). This decreasing trend was consistent with the trends of mass concentrations of $PM_{2.5}$
and $SO_2$ (Figure 2b & 2c), which were attributed to the implementation of the Clean Air Action
since 2013 in China (Zheng et al., 2018). As mentioned earlier (Section 2.5), the nationwide
reduction of anthropogenic mercury emissions should be largely responsible for the significant
decrease in GEM concentration observed at the YRD regional background site.
Seasonal average GEM concentrations decreased from 3.62 ng/m$^3$ to 2.17 ng/m$^3$ with a rate of
-0.37 ng m$^{-3}$ yr$^{-1}$ in spring, from 2.89 ng/m$^3$ to 1.98 ng/m$^3$ with a rate of -0.26 ng m$^{-3}$ yr$^{-1}$ in summer,
from 2.62 ng/m$^3$ to 1.94 ng/m$^3$ with a rate of -0.22 ng m$^{-3}$ yr$^{-1}$ in autumn, and from 2.91 ng/m$^3$ to
1.82 ng/m$^3$ with a rate of -0.35 ng m$^{-3}$ yr$^{-1}$ in winter (Figure 3). The decreasing rates of GEM were
~30% lower in the warm seasons than the cold seasons. Considering that seasonal variations of
anthropogenic emission are minimum, the different seasonal decreasing rates of GEM should be
mostly caused by the seasonal-dependent emission amounts from natural sources, knowing that
natural emissions are controlled by solar radiation and temperature, among other factors (Howard
and Edwards, 2018;Pannu et al., 2014;Mason, 2009).

**3.1.2 Impact of temperature on ambient gaseous elemental mercury**

In a previous study we showed that GEM concentrations tended to rise with increasing
temperature in the YRD region, which was considered to be the effect of temperature-dependent
emission amounts from natural surfaces (Qin et al., 2019). Here, to qualitatively investigate the role
of natural surface emissions on ambient GEM concentration, diurnal profiles of the bi-hourly GEM
concentration and temperature are exhibited in Figure 4. If looking at the whole year data together,
moderate to high correlations were seen between the diurnal variations of GEM and temperature in
2016, 2017, and 2018 with $R^2$ being 0.30 to 0.86, except in 2015 with little correlation with $R^2$ being
only 0.03 (Figure 4a-4d). The maximum GEM concentrations generally appeared around 10AM -
14PM, mostly coincided with daily peak temperature. These findings provided strong evidence of
temperature-dependent GEM sources.
Due to the large differences in ambient temperature between warm (from June to November)
and cold (from December to May) seasons in the YRD region, the effects of temperature-dependent
GEM sources on the ambient GEM concentrations should be different in different seasons. As
expected, high correlations between GEM concentration and temperature were found in the warm





seasons with $R^2$ being in the range of 0.15 to 0.87 (Figures 4e-4h), while nearly no correlations in
the cold seasons (Figures 4i-4l). Thus, the influence of natural surface emissions on ambient GEM
concentration was important in the warm seasons, but may not be the case in the cold seasons. The
seasonal bivariate polar plots of GEM showed that high GEM concentrations were associated
frequently with air flows from the south and southwest and occasionally with those from the north,
particularly in summer (Figure S1). This was consistent with the findings in previous studies which
showed stronger natural surface emissions in South and Southwest China than North China (Wang
et al., 2016;Feng et al., 2005;Wang et al., 2006;Sommar et al., 2016). Hence, in the context of
significant reduction of anthropogenic mercury emission in China, especially in North China (Liu
et al., 2019b), natural surface emissions significantly impacted the ambient GEM concentrations at
this sampling site.

**3.2 Quantify the contributions from natural surface emissions to ambient gaseous elemental**

**mercury**

**3.2.1 Development of the approach**

A method is developed below for quantifying the contributions of GEM emissions from natural
surfaces to ambient GEM concentrations through application of the PMF model by introducing
specific variables related to natural surface emissions as traces. The first step is to identify what
variables are directly or indirectly related to the natural surface emissions of GEM. Temperature is
certainly a dominant one as has been demonstrated in existing soil-air fluxes studies of mercury
(Wang et al., 2014b;Zhu et al., 2016;Poissant and Casimir, 1998). The formation pathways of $Hg^0$
in soil are all related to temperature, an empirical rule suggests that a 10℃ temperature increase
doubles the rates for chemical reaction near room temperature, which has been proven to be
applicable to $Hg^{II}$ reduction in boreal soil (Moore and Carpi, 2005;Quinones and Anthony,
2011;Wang et al., 2016;Pannu et al., 2014). Discussions in Section 3.1.2 also suggested temperature
as a potentially useful tracer for predicting natural surface emissions of GEM. A second candidate
of tracers could be ambient $NH_3$ concentration because soil emissions of GEM and $NH_3$, both of
which are temperature-dependent, are treated in a similar way in air-quality modeling studies
(Wright and Zhang, 2015;Zhang et al., 2010). The third potential tracer could be $O_3$ concentration
because high temperature can promote the formation of $O_3$ (Kerr et al., 2019;Kerr and Waugh,





2018;Schnell and Prather, 2017). As shown in Figure S2, the mean diurnal variations of GEM
concentration highly correlated with ambient temperature as well as $NH_3$ and $O_3$ concentrations.
From this perspective, $NH_3$ and $O_3$ can be regarded as indirect proxies for the natural surface
emissions of GEM. In a previous study we have applied principal component analysis for source
apportionment of mercury in this area, and the source factor with high loadings for temperature,
$NH_3$, and $O_3$ was interpreted as natural surface emissions of GEM (Qin et al., 2019).

252        Hence, in this study, we included the data of temperature, $NH_3$, and $O_3$ into the PMF model to

apportion the sources of GEM. As shown in Figures S3-S18, the source apportionment results for
all the seasons of 2015-2018 all resolved a similar factor with high loadings of temperature and $O_3$
and moderate loadings of $NH_3$ and GEM. This factor was thought to be the natural surface emission
sources of mercury. As for the other resolved factors, the factor with high loadings of V and Ni
evidently represented shipping emissions. The factor with high loadings of Ca was assigned to
cement production. Moderate loadings of multiple species including Cr, Mn, and Fe was found in
one factor which were identified as iron and steel production. The factor with high loadings of NO
was identified as vehicle emissions. And the last factor was identified as coal combustion due to the
high loadings of As and Se, and moderate contributions of Pb and $SO_4^{2-}$.

262        In order to verify the PMF modeling results, we first examined the PMF model performance.

Table S2 shows the coefficient of determination ($R^2$) for GEM according to the observation-
prediction scatter plots (Figure S20-S23). The $R^2$ values ranged from 0.37 to 0.89, suggesting an
acceptable model performance. Figure S24-S27 display the time series of observed and predicted
GEM concentrations from 2015-2018, which revealed that, except for a few extremely high
observation values, the model can relatively well reproduce the observed GEM concentration on an
hourly basis. To further verify the reliability of the resolved factors, the correlations between the
mass contributions of all factors to GEM and temperature were examined on the basis of diurnal
profiles. As shown in Figure S19, positive correlation was only found between the natural surface
emission factor and temperature while the other resolved factors (i.e. vehicle emission, coal
combustion, shipping activities, cement production, and iron and steel production) did not show this
relationship. This further corroborated that by using temperature, $NH_3$, and $O_3$ as tracers, the natural
surface emissions of GEM can be identified and quantified.



### 3.2.2 Increasing contributions from natural surface emissions to ambient gaseous elemental mercury

Figure 5 summarizes the contributions of natural surface emissions and anthropogenic emissions to GEM on seasonal basis from 2015 – 2018. The contributions of natural surface emissions to GEM were ~40% higher in summer ($1.09\pm0.58$ ng/m$^3$) than winter ($0.78\pm0.54$ ng/m$^3$). Besides, the contributions of natural surface emissions to GEM exhibited an upward trend, e.g., increased from 32% to 52% in spring, 39% to 63% in summer, 41% to 53% in autumn, and 32% to 43% in winter, from 2015-2018 (Figure 5). In contrast, the contributions from anthropogenic sources to GEM showed a downward trend, of which the decreased contribution from coal combustion accounted the most. Coal combustion has been widely regarded as the dominant anthropogenic source of mercury emissions at the global scale, and China is known as the largest coal producer and consumer in the world (Zhang et al., 2012;Wu et al., 2006). Since 2013, a series of key air pollution control measures have been applied in China to reduce the emission of air pollutants (Zheng et al., 2018). YRD regions also took actions by regulating on the amount of coal consumption, promoting renewable energy development and so on (Zheng et al., 2016). Hence, the decreased contribution of coal combustion was attributed to the implementation of aggressive air pollutant control measures in China in recent years, which subsequently led to an increase in the relative contribution of natural surface emissions to GEM.

The absolute GEM concentrations contributed by both natural surface emissions and anthropogenic emissions can be extracted from the PMF modeling results. Figure 6 exhibits the monthly and yearly profiles from 2015 to 2018. Strong seasonal cycles of GEM contributed by natural surface emissions were seen, corresponding to the seasonal pattern of ambient temperature (Figure 6g) and the simulated monthly Hg fluxes from natural surface emissions in China (Wang et al., 2016). The annual GEM concentration contributed by natural surface emissions was estimated to be $0.94\pm0.57$ ng/m$^3$, $1.01\pm0.63$ ng/m$^3$, $1.00\pm0.62$ ng/m$^3$, and $0.90\pm0.48$ ng/m$^3$ from 2015 to 2018, respectively (Figure 6a & 6b), which almost remained unchanged. This could be mainly explained by the little variation of annual temperature (Fig. 6h) and wind pattern from 2015 to 2018 (Fig. S28). On the contrary, the annual GEM concentration contributed by anthropogenic emissions was estimated to be $1.67\pm1.06$ ng/m$^3$, $1.51\pm0.77$ ng/m$^3$, $1.38\pm1.02$ ng/m$^3$, and $0.80\pm0.63$ ng/m$^3$ from 2015 to 2018, respectively, showing an obvious decreasing trend (Figure 6c & 6d). It was





noted that the GEM concentration contributed by anthropogenic emissions dropped the most from
2017 to 2018 with a rate of around 40%. By referring to the Table S3, $SO_2$ and CO also decreased
significantly of about 35% and 18%. As $SO_2$ and CO were the main primary gaseous pollutants
emitted from fuel combustions, their sharp decreases indicated the significant reduction of
anthropogenic emissions which was probably responsible for large drop of GEM from 2017 to 2018.
Overall, the relative contribution of natural surface emissions to ambient GEM was on the rise, e.g.,
from 36% in 2015 to 53% in 2018 on annual average (Figures 6e & 6f).

**4. Conclusions and Implications**

315       Through a four-year continuous measurement of GEM in the suburbs of Shanghai, a clear

decreasing trend was observed with the rate of -0.32±0.07 ng m$^{-3}$ yr$^{-1}$, which was mainly due to the
reduction of anthropogenic mercury emissions. The lower decreasing rate in warm seasons than in
cold seasons and the high correlation between GEM concentrations and temperature suggested that
natural surface emissions significantly impacted the GEM concentrations. By demonstrating that
temperature, $O_3$, and $NH_3$ can well serve as tracers of natural surface mercury emissions,
distinguishing natural vs. anthropogenic contributions to GEM was doable by introducing these
tracers into the PMF model. The results indicated that the contribution from anthropogenic mercury
emissions was declining, especially from coal combustion. The annual absolute contributions of
natural surface emissions were in the range of 0.90-1.01 ng/m$^3$, and the relative contribution of
natural surface emissions to GEM increasing form 36% in 2015 to 53% in 2018.

326       Measurements of GEM and other pollutants in a regional background area in Eastern China

demonstrated the effectiveness of emission control policies in this and surrounding regions in China
in recent years. The decreasing contributions from anthropogenic sources and the relatively stable
contributions from natural surface emissions to the ambient GEM have resulted in the relative
contributions of natural surface emissions surpassing those of anthropogenic emissions in more
recent years. This trend will likely continue for some years considering the current pollution levels
in China which needs further pollution abatement. This implies that even though the anthropogenic
emissions of mercury would continue to decrease, the legacy mercury in the natural surfaces will
continue to emit steadily for a long period of time. In addition, the natural release of mercury could
be enhanced under climate warming scenario. Hence, the atmospheric mercury concentration in



YRD or other parts of China will remain at relatively high levels in the near future, which brings
big challenges to China's policies on mercury emissions reduction. The methodology developed in
the present study could also shed some light on source apportionment of atmospheric mercury in
the other regions of the world, and has potential for improving emission databases from natural
surfaces where ambient GEM and auxiliary data are available.

**Acknowledgments**
The authors acknowledge support of the National Key R&D Program of China (2018YFC0213105),
the National Natural Science Foundation of China (91644105, 21777029), and the Natural Science
Foundation of Shanghai (18230722600,19ZR1421100).

**Author contribution**
X.Q. and K. H. designed this study. X.Q. performed measurements and data analysis. X.W., Q.F.,
Q.Z., Y.L., and J.H. performed data collection. X.Q., L.Z., K.H., and C.D. wrote the paper. All have
commented and reviewed the paper.

**Competing interests**
The authors declare that they have no conflict of interest.

**Data availability**
All data used in this study can be requested from K.H. (huangkan@fudan.edu.cn).

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



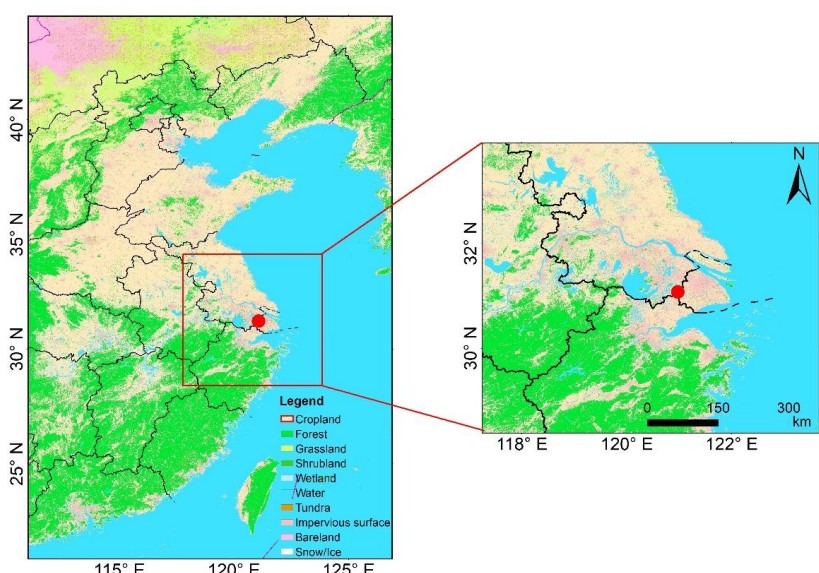

Figure 1. The location of the Dianshan Lake (DSL) site in Shanghai, China. Different colors in the
map represent different land cover types.


Figure 2. Monthly and annual variations of (a) GEM, (b) $PM_{2.5}$, and (c) $SO_2$ concentrations from
2015 to 2018.


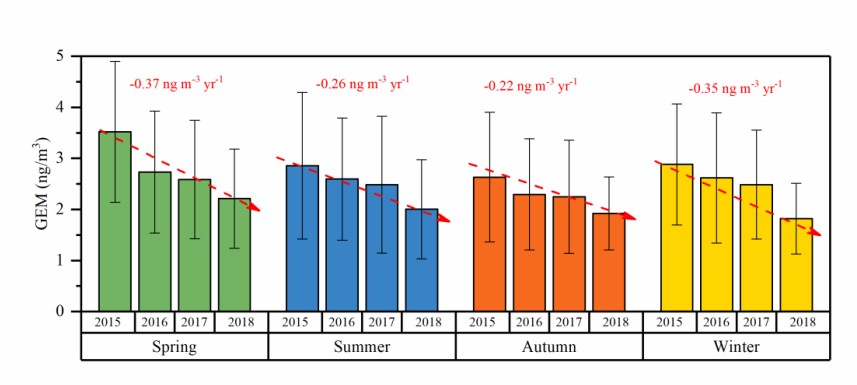

Figure 3. Seasonal variations of GEM concentrations from 2015 to 2018. The variation rates of
GEM for each season are also shown in the figure.


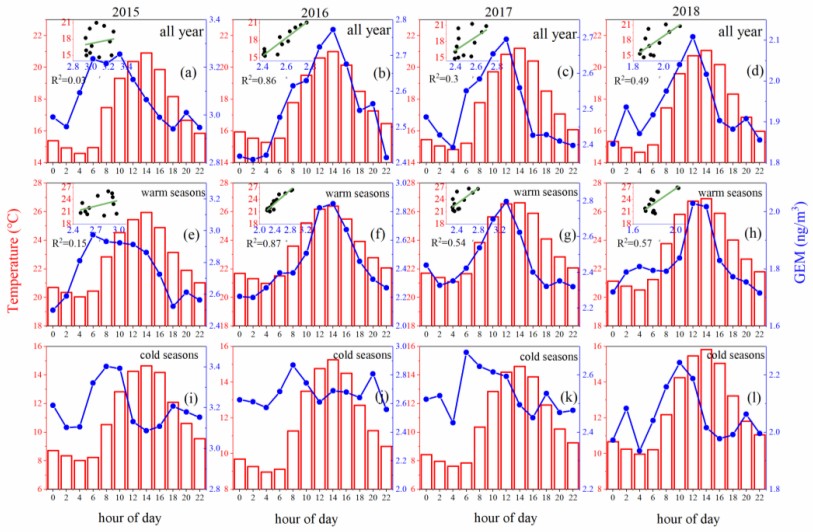

Figure 4. Diurnal patterns of bi-hourly GEM concentrations and temperature for the whole year (a-
d), warm seasons (e-h), and cold seasons (i-l) during 2015 – 2018, respectively. The linear
correlations between GEM and temperature are inserted as inner figures.





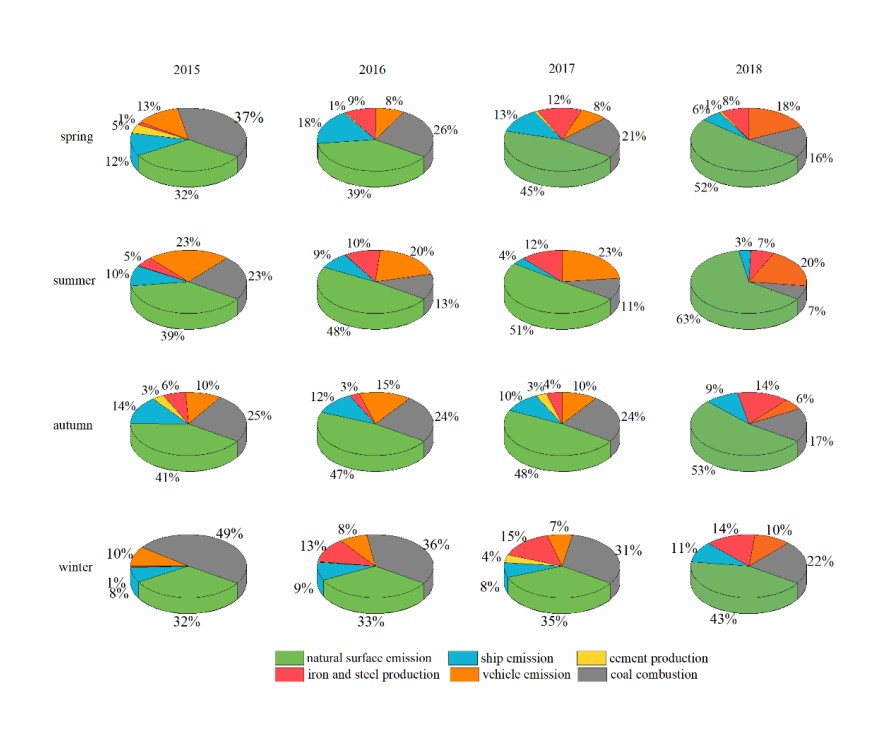


Figure 5. Contributions of natural surface emissions and anthropogenic sources to atmospheric
GEM in the four seasons during 2015 – 2018.


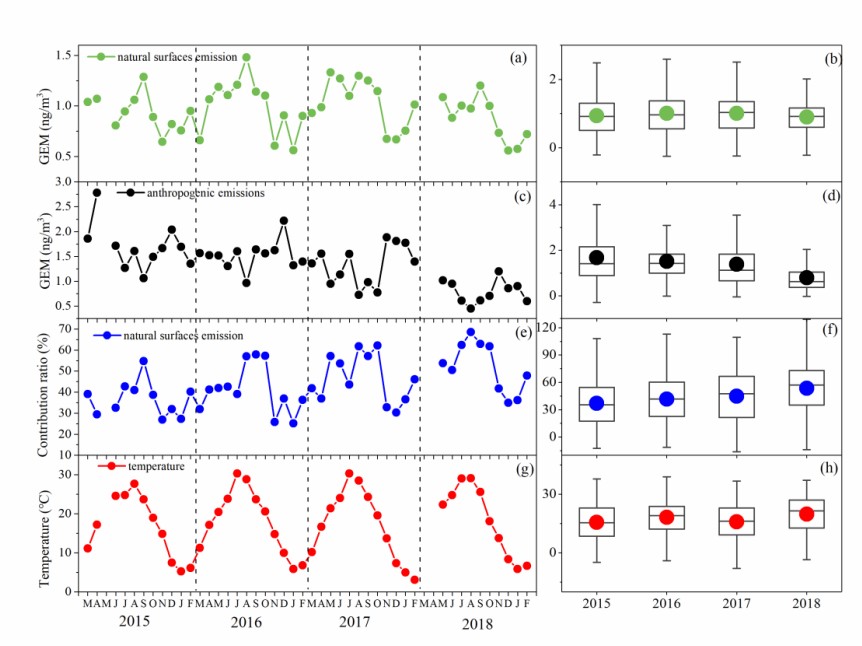

Figure 6. The monthly and annual GEM concentrations contributed by natural surface emissions (a-b) and anthropogenic emissions (c-d) from 2015 to 2018. (e-f) The monthly and annual contribution of natural surface emissions to GEM concentrations from 2015 to 2018. (g-h) The corresponding ambient temperature from 2015 to 2018.