# Peer review of "Assessing contributions of natural surface and anthropogenic emissions to"

_Atmospheric Chemistry and Physics, 2020_

## Referee Comment (RC1) · Anonymous Referee #1 · 1 Jun 2020

The manuscript entitled "Assessing contributions of natural surface and 1 anthropogenic emissions to atmospheric mercury in a fast developing region of Eastern China from 2015 to 2018", investigated the temporal variations of GEM, and developed a receptor model based method to quantify the contribution of natural surface mercury emission. The quantification of emission sources are significant to understand global mercury cycle. The development of the receptor model is one significant output of this study. However, the approach and the results is doubtful. It is true that when temperature increase, we can observe high GEM and $NH_3$ emissions from natural sources. $O_3$

is a typical secondary pollutant formed from VOCs and NOx, which mainly originates from photochemical reactions of anthropogenic pollutants and is impacted by temperature. The increase of temperature can also promote the generation of O3 as well. But the simultaneous changes of these three are not entirely the contribution of natural source emissions. Take a simple example. Both NH3 and mercury can participate between gas and particle. The increase of temperature will promote the generation of both NH3 and GEM. In addition, high temperature in summer generally promote the generation of O3. Thus, the simultaneous increase of NH3, O3, and GEM may occur due to atmospheric reaction process. Therefore, using O3 and NH3 as tracers of the natural emission of GEM will introduce a relative large uncertainty. The problem is that we do know how large the uncertainty will be, because we cannot exhaust this kind of examples considering the variable sources and generation pathways of these three air pollutants and the complicated impact from temperature. The results are also confusing. The author stated that "As for the other resolved factors, …. of Pb and SO42-"(Line 255-261). The explanation of the factors is too arbitrary and lacks enough support. For example, the authors pointed out that the factor with high loadings of Ca was assigned to cement production. However, there are several anthropogenic Ca emission sources if the authors investigated the heavy metal emission inventory, such as the ferrous metal smelting. Ferrous metal smelting is also one significant emission sources around Shanghai. From this aspect, the anthropogenic sources resolved by using the developed model can not be supported by the emission inventory. Due to the question of current receptor model and their definition of different factors, I think the authors need to carefully verify their results or use other source resolution methods to determine the sources.

---

## Referee Comment (RC2) · Anonymous Referee #2 · 7 Jun 2020

**Reviewer's Comments**

**General comment**

This study has analyzed data from multiyear measurements of the gaseous elemental mercury (GEM) concentration at a regional background site in eastern China and quantified the contribution of natural surface emission to GEM using the positive matrix factorization (PMF) model. The long-term observation data are valuable and the topic is of broad interests.

My major concern is the robustness of the PMF results. To what extent should we believe these results? The results need further verification. Figure 5 is one of the most important yields from this study. Suppose Figure 5 is basically correct, we can draw some important conclusions from this figure:

(1) Although cement production is believed to be one of the most important emission sources in China, it seems to contribute very little to GEM at this site. Could this be true?

(2) The current Hg emission inventories haven't considered ship emissions, but this emission source should be considered in the Hg emission inventory development, especially for coastal areas. This could be a very important finding if it is true.

(3) Iron and steel production has a large contribution to GEM concentration as well. Is this site under the influence of many large iron and steel plants (e.g., Baogang)?

If the contributions from different anthropogenic sources could be verified to some extent, it would be much easier for the readers to believe the contribution from natural sources. One possible approach for the verification that I can think of is to use the PSCF model to identify the potential GEM source regions from 2015 to 2018. If the key source regions for the monitoring site are consistent with the above conclusions (e.g., do not have many cement plants; have potential ship emissions from the seas or the rivers; have many iron and steel production activities; etc.), the robustness of the PMF model could be verified.

Overall, I think this manuscript is worth publishing on Atmospheric Chemistry and Physics after major revision.

**Specific comments**

1. Lines 47–48: It should be "non-ferrous metal smelters" instead of "non-ferrous smelters".

2. Section 2.2: How many valid GEM data were included in the analysis?

3. Lines 195–196: This statement is not accurate and lacks evidence. Some of the anthropogenic emission sources vary significantly from season to season. For example, coal combustion for residential use has a much higher level in winter.

4. Lines 208–209: The p values for all the correlations should be given here. Have the authors investigated the correlations between GEM and solar radiation? Solar radiation and temperature could have collinearity to a certain extent. It is possible that the diurnal GEM trend has a more significant correlation with solar radiation. Solar radiation is related to the photoreduction process of Hg in soil, which could be the major natural GEM source in the study area.

5. Lines 241–245: The choices of $NH_3$ and $O_3$ as tracers should be more carefully examined. These two tracers are not directly linked to natural emission sources, but indirectly through temperature. If temperature is already chosen as a tracer for PMF and $NH_3$ and $O_3$ are only linked to natural sources through temperature, what is the point of choosing $NH_3$ and $O_3$? The authors should pay attention to the other links between $NH_3$/$O_3$ and natural sources. Say the links through solar radiation, land surface type, and so on. Moreover, the PMF method usually chooses primary air pollutants as tracers, e.g., VOC species profiles, ions on particles, heavy metal profiles, etc. Secondary air pollutants, such as $O_3$, are usually avoided to be used as a tracer for PMF, because all the coefficients resulting from the PMF model need to be positive while it is not always the case for secondary air pollutants like $O_3$, not to mention that $O_3$ and GEM are potentially not independent variables. $O_3$ might act as an oxidizer for GEM under certain conditions (e.g., high humidity),

although this mechanism is not clear so far. Therefore, the authors should either remove $O_3$ as a tracer or explain why in this case $O_3$ is applicable from PMF.

---

## Author Comment (AC2) · 22 Jul 2020

**Response to reviewer's comments**

Anonymous Referee #2:

Major comments:

This study has analyzed data from multiyear measurements of the gaseous elemental mercury (GEM) concentration at a regional background site in eastern China and quantified the contribution of natural surface emission to GEM using the positive matrix factorization (PMF) model. The long-term observation data are valuable and the topic is of broad interests.

My major concern is the robustness of the PMF results. To what extent should we believe these results? The results need further verification. Figure 5 is one of the most important yields from this study. Suppose Figure 5 is basically correct, we can draw some important conclusions from this figure:

(1) Although cement production is believed to be one of the most important emission sources in China, it seems to contribute very little to GEM at this site. Could this be true?

(2) The current Hg emission inventories haven't considered ship emissions, but this emission source should be considered in the Hg emission inventory development, especially for coastal areas. This could be a very important finding if it is true.

(3) Iron and steel production has a large contribution to GEM concentration as well. Is this site under the influence of many large iron and steel plants (e.g., Baogang)?

If the contributions from different anthropogenic sources could be verified to some extent, it would be much easier for the readers to believe the contribution from natural sources. One possible approach for the verification that I can think of is to use the PSCF model to identify the potential GEM source regions from 2015 to 2018. If the key source regions for the monitoring site are consistent with the above conclusions (e.g., do not have many cement plants; have potential ship emissions from the seas or the rivers; have many iron and steel production activities; etc.), the robustness of the PMF model could be verified.

Overall, I think this manuscript is worth publishing on Atmospheric Chemistry and Physics after major revision.

We sincerely thank for the reviewer's in-depth comments and helpful suggestions on this manuscript. Based on the specific comments, we have responded to all the comments point-by-point and made corresponding changes in the manuscript as highlighted in red color. The reviewer has raised a number of issues and we quite agree. We feel the substantial revisions based on the reviewer's comments have greatly improved the quality of this manuscript. Please check the detailed responses to all the comments as below.

Specific comments:

1. Although cement production is believed to be one of the most important emission sources in China, it seems to contribute very little to GEM at this site. Could this be true?

Response: We agree with the reviewer that cement production is one of the most important mercury sources in China. According to the emission inventories, the annual GEM emission from cement production in the YRD region is around 2.3 tons/year, accounting for about 13% of its total anthropogenic emissions (Tang et al., 2018). By considering the natural sources of GEM (Zhu et al., 2016), the contribution of cement production to total GEM emissions should be lower than 13%. In this study, the seasonal contribution of cement production to the ambient GEM was estimated to be in the range of 2% - 10% at the study site. Hence, the PMF modeling results were generally consistent with the emission inventories.

2. The current Hg emission inventories haven't considered ship emissions, but this emission source should be considered in the Hg emission inventory development, especially for coastal areas. This could be a very important finding if it is true.

Response: Thanks for the comment. Shipping emissions are indeed important sources of air pollutants for the coastal areas, especially the East China Sea (Liu et al., 2017;Fan et al., 2016). However, as current Hg emission inventories haven't considered ship emissions as the reviewer mentioned, it is hard to verify the results of this study against the emission source data. Instead, we plotted the PSCF maps of GEM contributed from shipping emissions extracted from the PMF modeling results as shown in the figure below. The results showed strong PSCF signals from the coastal and oceanic areas, indicating the shipping factor resolved in this study is valid. Again, this should be verified when shipping mercury emission inventory is available in the future.

3. Iron and steel production has a large contribution to GEM concentration as well. Is this site under the influence of many large iron and steel plants (e.g., Baogang)?

Response: The figure below exhibit the geographical distribution of point sources in 2017 in China (Liu et al., 2019), which show that there are indeed many large iron and steel sites around our site (e.g., Baogang, Nangang, and Hanggang). According to the recent emission inventories, the contribution of iron and steel production accounts for about 7% of total anthropogenic GEM emissions (Tang et al., 2018). The seasonal contribution of iron and steel production to GEM ranged from 1% to 17% from 2015 to 2018 according to the PMF results. We believe that this site is under the influence of many large iron and steel plants.

---

## Author Response (AR1)

**Response to reviewer's comments**

Anonymous Referee #1:

Major comments:

The manuscript entitled "Assessing contributions of natural surface and anthropogenic emissions to atmospheric mercury in a fast developing region of Eastern China from 2015 to 2018", investigated the temporal variations of GEM, and developed a receptor model based method to quantify the contribution of natural surface mercury emission. The quantification of emission sources are significant to understand global mercury cycle. The development of the receptor model is one significant output of this study. However, the approach and the results is doubtful.

We sincerely thank for the reviewer's in-depth comments and helpful suggestions on this manuscript. Based on the specific comments, we have responded to all the comments point-by-point and made corresponding changes in the manuscript as highlighted in red color. The reviewer has raised a number of issues and we quite agree. We feel the substantial revisions based on the reviewer's comments have greatly improved the quality of this manuscript. Please check the detailed responses to all the comments as below.

Specific comments:

1. It is true that when temperature increase, we can observe high GEM and $NH_3$ emissions from natural sources. $O_3$ is a typical secondary pollutant formed from VOCs and NOx, which mainly originates from photochemical reactions of anthropogenic pollutants and is impacted by temperature. The increase of temperature can also promote the generation of $O_3$ as well. But the simultaneous changes of these three are not entirely the contribution of natural source emissions. Take a simple example. Both $NH_3$ and mercury can participate between gas and particle. The increase of temperature will promote the generation of both NH3 and GEM. In addition, high temperature in summer generally promote the generation of $O_3$. Thus, the simultaneous increase of $NH_3$, $O_3$, and GEM may occur due to atmospheric reaction process. Therefore, using $O_3$ and $NH_3$ as tracers of the natural emission of GEM will introduce a relative large uncertainty. The problem is that we do know how large the uncertainty will be, because we cannot exhaust this kind of examples considering the variable sources and generation pathways of these three air pollutants and the complicated impact from temperature

Response: Thanks a lot for the reviewer's insightful suggestion. We agree that $O_3$ is not suitable as the tracer of natural emission as it is a typical secondary pollutant and shouldn't be used as a tracer for PMF modeling. Hence, we have removed $O_3$ and re-run the PMF model for the whole multi-year dataset. The new modeling results are shown in the following figures. We found that after the removal of $O_3$, the contributions of natural and anthropogenic sources to GEM from 2015 to 2018 didn't change much, hence the major conclusion hasn't been affected. In general, the contributions of natural sources to GEM increased slightly. For example, before removing $O_3$, the relative contribution of natural surface emissions to GEM increase from 36%

in 2015 to 53% in 2018. After removing O$_3$, its contribution increases from 41% in 2015 to 57% in 2018. In the revision, we replace Figure 5 and Figure 6 with the following two figures, and modified the corresponding specific contribution values.

[Figure]

Contributions of natural surface emissions and anthropogenic sources to atmospheric GEM in the four seasons during 2015 – 2018.

[Figure]

The monthly and annual GEM concentrations contributed by natural surface emissions (a-b) and anthropogenic emissions (c-d) from 2015 to 2018. (e-f) The monthly and annual contribution of natural surface emissions to GEM concentrations from 2015 to 2018. (g-h) The corresponding ambient temperature from 2015 to 2018.

2. The results are also confusing. The author stated that "As for the other resolved factors, . . .. of Pb and $SO_4^{2-}$" (Line 255-261). The explanation of the factors is too arbitrary and lacks enough support.

Response: We agree with the reviewer that the explanation of the factors is not sufficient. In the revision, we revised the sentences as " As for the other resolved factors, the factor with high loadings of V and Ni evidently represented shipping emissions, because Ni and V have been considered as typical tracers of heavy oil combustion which has been commonly used in marine vessels (Viana et al., 2009). The factor with high loading of Ca was assigned to cement production as the raw materials used in cement production contain a large amount of calcium compounds. Moderate loadings of multiple species including Cr, Mn, and Fe were found in one factor which was identified as iron and steel production. The factor with high loading of NO was identified as vehicle emissions, as the major source of NOx in the YRD region is mobile oil combustion (Tang et al., 2018). And the last factor was identified as coal combustion due to the high loadings of As and Se, and moderate contributions from Pb and $SO_4^{2-}$. As, Se, and Pb were all typical tracers of coal combustion and the precursor of $SO_4^{2-}$ (i.e. $SO_2$) also mainly derived from coal combustion."

3. For example, the authors pointed out that the factor with high loadings of Ca was assigned to cement production. However, there are several anthropogenic Ca emission sources if the authors investigated the heavy metal emission inventory, such as the ferrous metal smelting. Ferrous metal smelting is also one significant emission sources around Shanghai. From this aspect, the anthropogenic sources resolved by using the developed model cannot be supported by the emission inventory.

Response: Thanks for the comment. According to the emissions inventories of China, non-ferrous metals smelting plants are mainly concentrated in Hunan, Yunnan, and Henan provinces (Liu et al., 2019). Hg emissions from non-ferrous metals smelting gradually decreased since 2004 in China, benefitting from the elimination of small-scale smelters and stringent $SO_2$ emission control measures (Wu et al., 2016). As for the YRD region, the recent emissions inventories show that the main emission sectors of GEM include coal-fired power plants, coal-fired industrial boilers, residential coal combustion, cement clinker production, iron and steel production, and mobile oil combustion, but very little from non-ferrous metal smelting (Tang et al., 2018). According to the emission inventories, the annual GEM emission from cement production in the YRD region is around 2.3 tons/year, accounting for about 13% of its total anthropogenic emissions (Tang et al., 2018). By considering the natural sources of GEM (Zhu et al., 2016) , the contribution of cement production to total GEM emissions should be lower than 13%. In this study, the seasonal contribution of cement production to the ambient GEM was estimated to be in the range of 2% - 10% at the study site. Hence, the PMF modeling results were generally consistent with the emission inventories.

4. Due to the question of current receptor model and their definition of different factors, I think the authors need to carefully verify their results or use other source resolution methods to determine the sources.

Response: Thanks for the comments and we do agree with the reviewer that the results should be carefully verified. In this regard, we have conducted more analysis to verify the results of PMF model from several aspects.

First, we verified whether the separation of natural and anthropogenic GEM was credible or not, which was also the main focus of this study. To achieve this, the relationship between particulate black carbon (BC) and GEM concentrations was investigated. On the one hand, BC mainly derived from various combustion processes, which were also the main anthropogenic sources of atmospheric mercury. On the other hand, BC was never introduced into the PMF modeling. As shown in the figure below, the observed total GEM concentrations and BC concentrations only showed weak correlations. This was mainly due to the fact that besides anthropogenic sources, natural sources also contributed significantly to GEM. As a comparison, anthropogenic GEM concentrations (extracted from PMF results) showed much stronger correlations with BC from 2015 to 2018. In addition, the time series of anthropogenic GEM concentrations generally varied consistently with CO (shown in the figure below), which is also a tracer of fuel combustion. This suggests that the PMF results are credible and the separation of anthropogenic and natural GEM has been successfully achieved.

[Figure]

The relationship between observed GEM and BC, anthropogenic GEM (extracted from PMF results) and BC during 2015 – 2018

[Figure]

Time series of anthropogenic GEM and CO concentrations

Furthermore, as shown in the figure below, we examine the time series of coal combustion GEM (extracted from PMF results) and observed $SO_2$ from 2015 to 2018. It is found that the trend of coal combustion GEM is basically consistent with that of $SO_2$, which indicates that the coal combustion factor resolved by PMF is credible.

[Figure]

Time series of coal combustion GEM and $SO_2$ concentrations

To verify the resolved shipping emission factor from PMF modeling, we using the PSCF model to identify the potential source regions of the shipping GEM (extracted from PMF results) from 2015 to 2018. As shown in the figures below, the potential source regions are mainly located over coastal and oceanic areas, which suggests that the shipping factor resolved by PMF is credible.

[Figure]

Potential source regions of shipping GEM from 2015 to 2018

In the revised manuscript, we have added a paragraph about the verification of PMF results as below.

"In addition, the relationship between particulate black carbon (BC) and GEM concentration was investigated. On the one hand, BC mainly derived from various combustion processes, which were also the main anthropogenic sources of atmospheric mercury. On the other hand, BC was never introduced into the PMF modeling. As shown in Figure 5, the observed total GEM and BC concentrations only showed weak correlations. This was mainly due to the fact that besides anthropogenic sources, natural sources also contributed significantly to GEM. As a comparison, anthropogenic GEM concentrations (extracted from PMF results) showed much better correlations with BC from 2015 to 2018. In addition, the time-series of anthropogenic GEM concentrations generally varied consistently with CO, which was also a tracer of fuel combustion (Figure S28). All the evidences above corroborated that by using temperature and NH$_3$ as tracers for PMF modeling, the separation of anthropogenic and natural GEM can be successfully achieved.

As for the specific anthropogenic mercury sources extracted from PMF results, Figure S29 shows that the time-series of coal combustion GEM also varied consistently with $SO_2$, indicating that the coal combustion factor resolved by PMF was credible. As shown in Figure S30, the potential source regions of shipping GEM were found mainly over coastal and oceanic areas, indicating the shipping factor resolved in this study was also valid. Figure S31 and Figure S32 show that the PSCF signals of cement production GEM were relatively weak in the YRD region, while there were substantial high PSCF signals for iron and steel production GEM in Eastern China. All the results above collectively confirmed that the PMF results were robust."

If the contributions from different anthropogenic sources could be verified to some extent, it would be much easier for the readers to believe the contribution from natural sources. One possible approach for the verification that I can think of is to use the PSCF model to identify the potential GEM source regions from 2015 to 2018. If the key source regions for the monitoring site are consistent with the above conclusions (e.g., do not have many cement plants; have potential ship emissions from the seas or the rivers; have many iron and steel production activities; etc.), the robustness of the PMF model could be verified.

Overall, I think this manuscript is worth publishing on Atmospheric Chemistry and Physics after major revision.

We sincerely thank for the reviewer's in-depth comments and helpful suggestions on this manuscript. Based on the specific comments, we have responded to all the comments point-by-point and made corresponding changes in the manuscript as highlighted in red color. The reviewer has raised a number of issues and we quite agree. We feel the substantial revisions based on the reviewer's comments have greatly improved the quality of this manuscript. Please check the detailed responses to all the comments as below.

Specific comments:

1. Although cement production is believed to be one of the most important emission sources in China, it seems to contribute very little to GEM at this site. Could this be true?

Response: We agree with the reviewer that cement production is one of the most important mercury sources in China. According to the emission inventories, the annual GEM emission from cement production in the YRD region is around 2.3 tons/year, accounting for about 13% of its total anthropogenic emissions (Tang et al., 2018). By considering the natural sources of GEM (Zhu et al., 2016), the contribution of cement production to total GEM emissions should be lower than 13%. In this study, the seasonal contribution of cement production to the ambient GEM was estimated to be in the range of 2% - 10% at the study site. Hence, the PMF modeling results were generally consistent with the emission inventories.

2. The current Hg emission inventories haven't considered ship emissions, but this emission source should be considered in the Hg emission inventory development, especially for coastal areas. This could be a very important finding if it is true.

Response: Thanks for the comment. Shipping emissions are indeed important sources of air pollutants for the coastal areas, especially the East China Sea (Liu et al., 2017;Fan et al., 2016). However, as current Hg emission inventories haven't considered ship emissions as the reviewer mentioned, it is hard to verify the results of this study against the emission source data.
Instead, we plotted the PSCF maps of GEM contributed from shipping emissions extracted from the PMF modeling results as shown in the figure below. The results showed strong PSCF signals from the coastal and oceanic areas, indicating the shipping factor resolved in this study is valid. Again, this should be verified when shipping mercury emission inventory is available in the future.

[Figure]

3. Iron and steel production has a large contribution to GEM concentration as well. Is this site under the influence of many large iron and steel plants (e.g., Baogang)?

Response: The figure below exhibit the geographical distribution of point sources in 2017 in China (Liu et al., 2019), which show that there are indeed many large iron and steel sites around our site (e.g., Baogang, Nangang, and Hanggang). According to the recent emission inventories, the contribution of iron and steel production accounts for about 7% of total anthropogenic GEM emissions (Tang et al., 2018). The seasonal contribution of iron and steel production to GEM ranged from 1% to 17% from 2015 to 2018 according to the PMF results. We believe that this site is under the influence of many large iron and steel plants.

[Figure]

4. One possible approach for the verification that I can think of is to use the PSCF model to identify the potential GEM source regions from 2015 to 2018. If the key source regions for the monitoring site are consistent with the above conclusions (e.g., do not have many cement plants; have potential ship emissions from the seas or the rivers; have many iron and steel production activities; etc.), the robustness of the PMF model could be verified.

Response: According to the reviewer's suggestion, we have identified the potential source regions of the PMF modeled GEM from cement production, iron and steel production, and shipping activities during 2015 - 2018, respectively. As shown in the figures below, the PSCF signals of GEM from cement production in the YRD region are relatively weak, while there are substantial high PSCF signals for iron and steel production GEM in Eastern China. As for GEM from the shipping sector, most high PSCF signals are from the coastal and oceanic areas. These results suggest that the PMF results in this study are credible.

[Figure]

Potential source regions of GEM from cement production during 2015 - 2018

[Figure]

Potential source regions of GEM from iron and steel production during 2015 - 2018

[Figure]

Potential source regions of GEM from shipping activities during 2015 - 2018

As Reviewer#1 also raised similar concerns, we added additional analysis about the verification of PMF results as below.

We verified whether the separation of natural and anthropogenic GEM was credible or not, which is also the main focus of this study. To achieve this, the relationship between particulate black carbon (BC) and GEM concentrations was investigated. On the one hand, BC mainly derives from various combustion processes, which are also the main anthropogenic sources of atmospheric mercury. On the other hand, BC was never introduced into the PMF modeling. As shown in the figure below, the observed total GEM concentrations and BC concentrations only showed weak correlations. This was mainly due to the fact that besides anthropogenic sources, natural sources also contributed significantly to GEM. As a comparison, anthropogenic GEM concentrations (extracted from PMF results) showed much stronger correlations with BC from 2015 to 2018. In addition, the time series of anthropogenic GEM concentrations generally varied consistently with CO (shown in the figure below), which is also a tracer of fuel combustion. This suggests that the PMF results are credible and the separation of anthropogenic and natural GEM has been successfully achieved.

[Figure]

The relationship between observed GEM and BC, anthropogenic GEM (extracted from PMF results) and BC during 2015 – 2018

[Figure]

Time series of anthropogenic GEM and CO concentrations

Furthermore, as shown in the figure below, we examine the time series of coal combustion GEM (extracted from PMF results) and observed $SO_2$ from 2015 to 2018. It is found that the trend of coal combustion GEM is basically consistent with that of $SO_2$, which indicates that the coal combustion factor resolved by PMF is credible.

[Figure]

Time series of coal combustion GEM and $SO_2$ concentrations

In the revised manuscript, we have added a paragraph about the verification of PMF results as below.

"In addition, the relationship between particulate black carbon (BC) and GEM concentration was investigated. On the one hand, BC mainly derived from various combustion processes, which were also the main anthropogenic sources of atmospheric mercury. On the other hand, BC was never introduced into the PMF modeling. As shown in Figure 5, the observed total GEM and BC concentrations only showed weak correlations. This was mainly due to the fact that besides anthropogenic sources, natural sources also contributed significantly to GEM. As a comparison, anthropogenic GEM concentrations (extracted from PMF results) showed much better correlations with BC from 2015 to 2018. In addition, the time-series of anthropogenic

GEM concentrations generally varied consistently with CO, which was also a tracer of fuel combustion (Figure S28). All the evidences above corroborated that by using temperature and NH₃ as tracers for PMF modeling, the separation of anthropogenic and natural GEM can be successfully achieved.

As for the specific anthropogenic mercury sources extracted from PMF results, Figure S29 shows that the time-series of coal combustion GEM also varied consistently with SO₂, indicating that the coal combustion factor resolved by PMF was credible. As shown in Figure S30, the potential source regions of shipping GEM were found mainly over coastal and oceanic areas, indicating the shipping factor resolved in this study was also valid. Figure S31 and Figure S32 show that the PSCF signals of cement production GEM were relatively weak in the YRD region, while there were substantial high PSCF signals for iron and steel production GEM in Eastern China. All the results above collectively confirmed that the PMF results were robust."

5. Lines 47–48: It should be "non-ferrous metal smelters" instead of "non-ferrous smelters".

Response: The statement "non-ferrous smelters" has been change as "non-ferrous metal smelters" in the revision.

6. Section 2.2: How many valid GEM data were included in the analysis?

Response: The sentence "In this study, the number of valid GEM data was 16266" has been added in the revision.

7. Lines 195–196: This statement is not accurate and lacks evidence. Some of the anthropogenic emission sources vary significantly from season to season. For example, coal combustion for residential use has a much higher level in winter.

Response: Thanks for pointing out this inaccurate statement. the sentence "Considering that seasonal variations of anthropogenic emission are minimum" has been revised as "Considering that the anthropogenic emissions were less temperature dependent, the different seasonal decreasing rates of GEM between the warm and cold seasons should be mostly caused by the seasonal-dependent emission amounts from natural sources" in the revision.

8. Lines 208–209: The p values for all the correlations should be given here.

Response: The p values for all the correlations have been added in the revision.

9. Have the authors investigated the correlations between GEM and solar radiation? Solar radiation and temperature could have collinearity to a certain extent. It is possible that the diurnal GEM trend has a more significant correlation with solar radiation. Solar radiation is related to the photoreduction process of Hg in soil, which could be the major natural GEM source in the study area.

Response: Thanks for this valuable suggestion. We quite agree with the reviewer that solar radiation is a key factor of the photoreduction process of Hg in soil and the diurnal GEM trend likely has a significant correlation with solar radiation. However, due to that solar radiation was not measured in this study, we cannot carry out the corresponding analysis. We will certainly consider the investigation of the relationship between GEM and solar radiation in the future field experiments.

10. Lines 241–245: The choices of NH3 and $O_3$ as tracers should be more carefully examined. These two tracers are not directly linked to natural emission sources, but indirectly through temperature. If temperature is already chosen as a tracer for PMF and NH3 and $O_3$ are only linked to natural sources through temperature, what is the point of choosing NH3 and $O_3$? The authors should pay attention to the other links between $NH_3/O_3$ and natural sources. Say the links through solar radiation, land surface type, and so on. Moreover, the PMF method usually chooses primary air pollutants as tracers, e.g., VOC species profiles, ions on particles, heavy metal profiles, etc. Secondary air pollutants, such as $O_3$, are usually avoided to be used as a tracer for PMF, because all the coefficients resulting from the PMF model need to be positive while it is not always the case for secondary air pollutants like $O_3$, not to mention that $O_3$ and GEM are potentially not independent variables. $O_3$ might act as an oxidizer for GEM under certain conditions (e.g., high humidity), although this mechanism is not clear so far. Therefore, the authors should either remove $O_3$ as a tracer or explain why in this case $O_3$ is applicable from PMF.

Response: After considering the reviewer's insightful suggestion, we agree that $O_3$ is not suitable as the tracer of natural emission and shouldn't be used as a tracer for PMF modeling. Hence, we have removed $O_3$ and re-run the PMF model for the whole multi-year dataset. The new modeling results are shown in the following figures. We found that after the removal of $O_3$, the contributions of natural and anthropogenic sources to GEM from 2015 to 2018 didn't change much, hence the major conclusion hasn't been affected. In general, the contributions of natural sources to GEM increased slightly. For example, before removing $O_3$, the relative contribution of natural surface emissions to GEM increase from 36% in 2015 to 53% in 2018. After removing $O_3$, its contribution increases from 41% in 2015 to 57% in 2018. In the revision, we replace Figure 5 and Figure 6 with the following two figures, and modified the corresponding specific contribution values.

[Figure]

Contributions of natural surface emissions and anthropogenic sources to atmospheric GEM in the four seasons during 2015 – 2018.

[Figure]

The monthly and annual GEM concentrations contributed by natural surface emissions (a-b) and anthropogenic emissions (c-d) from 2015 to 2018. (e-f) The monthly and annual contribution of natural surface emissions to GEM concentrations from 2015 to 2018. (g-h) The corresponding ambient temperature from 2015 to 2018.

[revised manuscript text omitted]

---

## Referee Report (RR1)

1. Why do you only remove $O_3$? Do you ever test the impact of $NH_3$?

2. The following, *natural emissions*, *nature surface emissions, and surface emissions*, seem refer to the same thing. Please be consistent throughout the whole text including in the figure. Moreover, re-emission is a significant source of atmospheric Hg, how do you consider its contribution?

3. Original comment 3: ***Ferrous metal smelting*** such as iron and steel smelting, instead of **non-ferrous metal smelting**, is one significant source of Ca emission. That is why I do not agree that *The factor with high loading of Ca was assigned to cement production as the raw materials used in cement production contain a large amount of calcium compounds.* The authors should notice that it is just one case. Actually, the results of Figure 6 is still under doubt. For example, ship and vehicle emissions are two large anthropogenic emission sources in Shanghai. One of the problems to this result is that the indicators used in PMF currently are not source-specific indicators and much depend on the subjective judgment of the authors. Under such situations, analyzing the uncertainty of the results is particularly important.

---

## Author Response (AR2)

**Response to reviewer's comments**

Anonymous Referee #1:

1. This manuscript has been greatly improved. Most of the reviewers' comments have been properly addressed in the authors' responses. I have one more comment for the manuscript: The authors could add some recommendations to the last part of the manuscript based on their findings. For example, ship emissions might be an important source of Hg which has been neglected in previous emission inventory studies. Field measurements of Hg emissions from ships are needed to verify this point. Moreover, this was one of the very few studies that apply PMF to atmospheric Hg source apportionment. What were the limitations of this PMF application? What kind of special attention should be paid to in the PMF analysis for Hg in the future?

Overall, I think this manuscript is ready for publication on Atmospheric Chemistry and Physics after minor revision.

Response: Thanks for the reviewer's valuable suggestions. we added the sentences "It has to be noted that according to our results, ship emission was identified to be an important source of mercury which hasn't been included in previous emission inventory studies. Mercury emission factors from shipping are lacking and field measurements will be needed to verify it. Moreover, we realize that the application of PMF in the source apportionment of atmospheric mercury sources has certain limitations. Source-specific indicators are preferred, however, most indicators used for source identification have multiple sources, which have added uncertainties to our results. We suggest that in future research, the application of the PMF model is limited to the separation of the natural and total anthropogenic sources of GEM, which has been proved to be credible."

**Response to reviewer's comments**

Anonymous Referee #2:

1. Why do you only remove $O_3$? Do you ever test the impact of $NH_3$?

Response: Thanks for the comment. As a secondary pollutant, the formation processes of $O_3$ in the atmosphere are very complicated, which is controlled by the relative emissions of NOx and VOCs in addition to solar radiation. In this regard, $O_3$ may not be a suitable input for PMF modeling and we removed it. In contrast, $NH_3$ mainly comes from primary emissions. One study at our sampling site have found the good correlation between $NH_3$ and temperature (Chang et al., 2019), and the soil emissions of GEM and $NH_3$ are treated in a similar way in air-quality modeling studies (Wright and Zhang, 2015;Zhang et al., 2010). Hence, we think that $NH_3$ can be regarded as an indirect tracer for the natural surface emissions of GEM.

2. The following, natural emissions, nature surface emissions, and surface emissions, seem refer to the same thing. Please be consistent throughout the whole text including in the figure. Moreover, re-emission is a significant source of atmospheric Hg, how do you consider its contribution?

Response: Thanks for the comment. We have carefully checked the full text and unified the term "natural surface emissions" to describe the natural sources of GEM in the revision. We agree with the reviewer that re-emission is a significant source of atmospheric mercury and its contribution has been included in the modeled GEM from natural surface emissions. However, it is difficult to separate the re-emission of deposited mercury and the emission of the original mercury in the natural surface by PMF only, which is also not the focus of this study.

3. Original comment 3: Ferrous metal smelting such as iron and steel smelting, instead of non-ferrous metal smelting, is one significant source of Ca emission. That is why I do not agree that The factor with high loading of Ca was assigned to cement production as the raw materials used in cement production contain a large amount of calcium compounds.

Response: We agree with the reviewer that ferrous metal smelting such as iron and steel smelting, cement production, and some other emission sectors all contribute to Ca emissions. The source profiles of those emission sectors can be similar at some extents. We did test to increase the number of PMF factors (e.g. 7, 8, and more), however, the separation of more detailed emission sources was not successful. Hence, the current PMF modeling framework may not well resolve more explicit emission source before introducing more specific tracers. This is the limitation of the PMF model if the selected indicator has multiple sources. We will try to improve it by introducing more unique tracers in the future works.

4. The authors should notice that it is just one case. Actually, the results of Figure 6 is still under doubt. For example, ship and vehicle emissions are two large anthropogenic emission sources in Shanghai. One of the problems to this result is that the indicators used in PMF

currently are not source-specific indicators and much depend on the subjective judgment of the authors. Under such situations, analyzing the uncertainty of the results is particularly important.

Response: We quite agree with the reviewer that most indicators used in PMF currently are not source-specific indicators and that contributed to the biggest uncertainties of the results. Currently, uncertainty analysis is not available via PMF. This is indeed the limitation of the PMF model.

In the conclusion section, we add the sentence "It has to be noted that according to our results, ship emission was identified to be an important source of mercury which hasn't been included in previous emission inventory studies. Mercury emission factors from shipping are lacking and field measurements will be needed to verify it. Moreover, we realize that the application of PMF in the source apportionment of atmospheric mercury sources has certain limitations. Source-specific indicators are preferred, however, most indicators used for source identification have multiple sources, which have added uncertainties to our results. We suggest that in future research, the application of the PMF model is limited to the separation of the natural and total anthropogenic sources of GEM, which has been proved to be credible."

**References:**

Chang, Y. H., Zou, Z., Zhang, Y. L., Deng, C. R., Hu, J. L., Shi, Z. H., Dore, A. J., and Collett, J. L.: Assessing Contributions of Agricultural and Nonagricultural Emissions to Atmospheric Ammonia in a Chinese Megacity, Environmental science & technology, 53, 1822-1833, 2019.

[revised manuscript text omitted]